# Myosteatosis Is Not Associated with Complications or Survival in HCC Patients Undergoing Trans Arterial Embolization

**DOI:** 10.3390/jcm12010262

**Published:** 2022-12-29

**Authors:** Chiara Masetti, Nicola Pugliese, Ludovica Lofino, Francesca Colapietro, Roberto Ceriani, Ana Lleo, Dario Poretti, Vittorio Pedicini, Stella De Nicola, Guido Torzilli, Lorenza Rimassa, Alessio Aghemo, Ezio Lanza

**Affiliations:** 1Division of Internal Medicine and Hepatology, Department of Gastroenterology, IRCCS Humanitas Research Hospital, Rozzano, 20089 Milan, Italy; 2Department of Biomedical Sciences, Humanitas University, Pieve Emanuele, 20072 Milan, Italy; 3Division of Interventional Radiology, IRCCS Humanitas Research Hospital, Rozzano, 20089 Milan, Italy; 4Division of Hepatobiliary and General Surgery, Department of Surgery, IRCCS Humanitas Research Hospital—IRCCS, Rozzano, 20089 Milan, Italy; 5Medical Oncology and Hematology Unit, Humanitas Cancer Center, IRCCS Humanitas Research Hospital, Rozzano, 20089 Milan, Italy

**Keywords:** hepatocellular carcinoma, sarcopenia, myosteatosis, trans-arterial embolization, locoregional therapy

## Abstract

Alterations in nutritional status, in particular sarcopenia, have been extensively associated with a poor prognosis in cirrhotic patients regardless of the etiology of liver disease. Less is known about the predictive value of myosteatosis, defined as pathological fat infiltration into the skeletal muscle. We retrospectively analyzed a cohort of 151 cirrhotic patients with unresectable hepatocellular carcinoma (HCC) who underwent their first trans-arterial embolization (TAE) between 1 March 2011 and 1 July 2019 at our Institution. Clinical and biochemical data were collected. Sarcopenia was assessed using the L3-SMI method while myosteatosis with a dedicated segmentation suite (3D Slicer), using a single slice at an axial plane located at L3 and calculating the IMAC (Intramuscular Adipose Tissue Content Index). The sex-specific cut-off values for defining myosteatosis were IMAC > −0.44 in males and >−0.31 in females. In our cohort, 115 (76%) patients were included in the myosteatosis group; 128 (85%) patients had a coexistent diagnosis of sarcopenia. Patients with myosteatosis were significantly older and showed higher BMI than patients without myosteatosis. In addition, male gender and alcoholic- or metabolic-related cirrhosis were most represented in the myosteatosis group. Myosteatosis was not associated with a different HCC burden, length of hospitalization, complication rate, and readmission in the first 30 days after discharge. Overall survival was not influenced by the presence of myosteatosis.

## 1. Introduction

Alterations in body composition have gained clinical relevance in recent years as they have been widely associated with adverse outcomes in a variety of medical conditions, including chronic liver disease (CLD) [1,2]. The importance of nutritional status and its assessment in these patients is further highlighted by the European Association for the Study of Liver (EASL) and American Association for the study of Liver Diseases (AASLD) guidelines on nutrition in CLD [3,4].

In addition to the loss of muscle mass and function, known as sarcopenia, qualitative changes in the structural composition of muscle, such as myosteatosis, have also been associated with worse clinical outcomes in different pathological conditions, especially cancer [5,6].

Myosteatosis is defined as the fatty infiltration of muscle, both in myocytes (intramyocellular fat) and in muscle fascia (intermuscular fat). It can be diagnosed with invasive techniques, such as muscle biopsy, or non-invasive ones such as computed tomography (CT), magnetic resonance imaging (MRI), and magnetic resonance spectroscopy (MRS) [7]. Most studies conducted on a large-scale population have diagnosed myosteatosis through CT scanning using the muscle radiation attenuation.

In recent years, several studies have focused on the role of myosteatosis in CLD, aiming to assess its impact on prognosis in patients with cirrhosis or in those undergoing orthotopic liver transplantation (OLT). Myosteatosis, similarly to sarcopenia, acts as a negative prognostic factor for morbidity, mortality, and adverse perioperative outcomes [8,9,10]. Whether these two conditions are directly linked, or, on the other hand, are independently associated with outcomes, is unknown.

On the other hand, data on the impact of myosteatosis on the outcome of patients with hepatocellular carcinoma (HCC) are currently very limited. Kaibori et al. reported a strong association of fat infiltration in skeletal muscle with inferior overall survival in patients suffering from HCC [11]. Furthermore, Meister et al. recently demonstrated the prognostic role of myosteatosis in a homogeneous cohort of HCC patients undergoing curative-intent surgery [12]. To date, there are no data on the impact of myosteatosis in patients with advanced HCC undergoing locoregional therapy.

By analyzing a large cohort of patients with HCC who underwent the first transarterial embolization (TAE) in a single center in Milan, and in whom the impact of sarcopenia was already reported, we were able to assess the impact of myosteatosis on overall survival, length of hospitalization, incidence of post-procedural complications, early rehospitalization, and its association with sarcopenia [13].

## 2. Materials and Methods

### 2.1. Design of the Study

This is a sub-study of a previously published study where we retrospectively included all consecutive patients with a diagnosis of HCC who underwent the first TAE from 1 March 2011 to 1 July 2019 at our Division of Hepatology [13]. Indication to TAE was given by a multidisciplinary team including hepatologists, hepato-biliary surgeons, oncologists, interventional radiologists, and radiation oncologists. All patients underwent a CT scan before being referred to TAE and they were followed up with CT imaging 1 month after the procedure. In case of disease recurrence or partial response to treatment, patients were re-discussed with a multidisciplinary team to decide the subsequent treatment; otherwise, a 3-month follow-up interval was established. Technical variations included microparticles TAE (P-TAE), microparticles plus cyanoacrylate glue TAE (G-TAE), and Lipiodol TAE (L-TAE). All procedures were performed by two senior operators and two junior interventional radiologists with at least 3 years’ experience in an angiographic suite (V5000 Philips Medical System, Amsterdam, The Netherlands), equipped with cone-beam CT (Siemens, Medical Solutions, Forchheim, Germany).

For patients meeting the inclusion criteria, a detailed clinical history was obtained, including past medical history, comorbidities, etiology of liver disease, number of HCC nodules, performance status, early complications rate, length of hospitalization, readmission rate within 30 days, and survival time from the first TAE. Biochemical data, such as serum ALT, AST, alpha-fetoprotein, platelet count, INR, albumin, and total and direct bilirubin were recorded, if available. Body mass index (BMI), MELD score, Child-Turcotte-Pugh score (CPT), albumin-bilirubin score (ALBI score), and BCLC stage were calculated for each patient.

### 2.2. Assessment of Myosteatosis

For the assessment of myosteatosis, a single cross-sectional CT image at the level of the third lumbar vertebra (L3) was used. The segmentation of skeletal muscle and adipose tissue was performed by the same researcher using the 3D Slicer software platform version 4.1 and BC module (https://www.slicer.org/ accessed on 15 December 2021) in a semiautomatic fashion. The muscle quality was defined by intramuscular adipose tissue content (IMAC), whose efficacy and feasibility has been widely validated [11,14]. IMAC was calculated by dividing the CT attenuation of the multifidus muscle (Hounsfield units) by that of subcutaneous adipose tissue (Hounsfield units). The value of IMAC is normalized to the value of subcutaneous fat individually; therefore, it is not influenced by the CT system or scanning condition. Higher IMAC denote a greater amount of fat infiltration in muscle tissue and, consequently, a worse quality of muscle. The sex-specific cut-off values for defining myosteatosis were IMAC > −0.44 in males and >−0.31 in females, respectively [11,14].

### 2.3. Statistical Analysis

All statistical analyses had been conducted using NCSS 10 statistical Software (NCSS). Continuous variables were expressed as mean ± standard deviation, while categorical variables were expressed as percentages. Univariate analysis was performed using Student T-test and Chi-square test for continuous and categorical variables, respectively. A value of *p* < 0.05 was considered statistically significant.

## 3. Results

### 3.1. Patients Characteristics

The main characteristics of the cohort have already been reported and are showed in Table 1. The majority of patients (116/151, 76.8%) were males, with a median age of 73.2 ± 9.3 years. Median BMI was 25.8 ± 4.9 kg/m^2^. Liver disease was caused by a viral etiology in nearly half of the enrolled patients, with HCV being represented in 45.3% of them and HBV in 5.3%. In 34 (22.7%) and 23 (15.3%) patients, the etiology was alcoholic or metabolic, respectively, while 11.4% had a rare cause of liver cirrhosis including hemochromatosis, primary biliary cholangitis, and autoimmune hepatitis. The majority of patients had well-compensated liver disease with preserved liver function tests: 82% of patients were in CPT class A, while 17.2% were in CPT class B, and only 0.8% had a decompensated liver disease in class C. The median MELD score was 10 ± 3, and 85.3% of patients had a coexistent diagnosis of sarcopenia. The ALBI score was available in 124 patients: 28 patients (22.6%) presented with Grade 1, 86 (69.3%) had Grade 2, while only 10 patients (8.1%) had Grade 3. BCLC stage was intermediate in the majority of patients (96, 63.5%), while 43 (28.5%) were considered early stage and 12 (8%) were in the very early stage. 

### 3.2. HCC Treatment

HCC was multifocal in 50% of patients, while 28% and 22% of patients presented with a monofocal or bifocal HCC, respectively. The mean diameter of the major lesion was 33 ± 22 mm. In total, 41.1% of patients had already received at least one treatment for HCC other than TAE. The mean length of hospitalization after TAE was 2 ± 1.7 days, with only 11 patients (7.3%) developing a complication after procedure and nine patients (6%) requiring rehospitalization in the first 30 days after discharge. We found that 27.1% of patients had a complete response after the first TAE, and the median overall survival of the cohort was 808 ± 673 days.

### 3.3. Prevalence of Myosteatosis and Its Impact on Safety and Efficacy of TAE

Of the total cohort, 115 patients had myosteatosis (76%) at CT examination and were included in the myosteatosis group, while the remaining 36 patients did not present with myosteatosis. Patients with myosteatosis were significantly older (74.8 ± 7.8 vs. 68.0 ± 11.7 years, *p* = 0.0001) and showed a higher BMI than patients without myosteatosis (26.6 ± 5.1 vs. 23.4 ± 3.3 kg/m^2^, *p* = 0.0006). Male gender was also associated with myosteatosis as the proportion of male patients was significantly higher in the myosteatosis group (80.9% vs. 63.9%, *p* = 0.035). The two groups showed no significant differences in terms of transaminases levels, albumin, bilirubin, INR, CPT class, and MELD score. Interestingly, the etiology of the underlying liver diseases was associated with the presence of myosteatosis, as there was a significantly higher proportion of patients with alcoholic or metabolic-related cirrhosis in the myosteatosis group (26.3% and 16.7% vs. 11.1% and 11.1%, respectively). Myosteatosis was not associated with sarcopenia defined by SMI at CT scan, as sarcopenia was equally represented in both groups (85.1% vs. 86.1%, *p* = 0.88).

Myosteatosis was not associated with different HCC burden since we did not observe any significant differences between the two groups regarding number of HCC nodules and diameter of the largest lesion. Length of hospitalization was similar in the two groups (2.1 ± 1.8 vs. 1.8 ± 1.3 days, *p* = 0.36), as were readmission rates in the first 30 days after discharge (6% vs. 5.5%, *p* = 0.90). Complication rates were numerically, albeit not statistically, higher in the myosteatosis group (7.8%, vs. 5.5% in patients without myosteatosis) (*p* = 0.64).

Overall survival was not influenced by myosteatosis as shown by the Kaplan-Meier curves (*p* = 0.94, Figure 1).

In the multivariate analysis, BMI, age, and male gender were significantly correlated with the presence of myosteatosis.

## 4. Discussion and Conclusions

Our study showed that myosteatosis, defined as a fatty infiltration of the muscle assessed by CT scanning, was highly prevalent (76%) in patients with HCC who received loco-regional treatment. The prevalence in our study was higher than what previously reported in other cohorts; indeed, in a large cohort of 678 patients with cirrhosis, myosteatosis was present in 52% of patients and was associated with significantly worse median survival, representing an independent predictor of mortality at Cox-regression analysis [8]. In that study, the presence of sarcopenia and myosteatosis was associated with a 1.5–2-fold higher risk of mortality compared to patients without muscular abnormalities. Since the presence of myosteatosis has been associated with more rapid fibrosis progression and higher values of liver stiffness, we could explain the higher prevalence reported in our cohort as the direct consequence of an analysis skewed toward patients with more advanced disease [15]. The median age of our cohort might also play a significant role in determining an increased prevalence of myosteatosis, as older age is associated with the presence of myosteatosis. This is the main finding reported by Gallagher and colleagues, who measured intermuscular adipose tissue (IMAT) in a large cohort of 338 patients of different races [16]. The authors found that IMAT increases significantly with age, independently of race group. Interestingly, age played a major role also in our cohort, as patients in the myosteatosis group were significantly older than patients without myosteatosis.

Interestingly, we could not find any impact of myosteatosis on overall survival following TAE, nor did we observe longer hospitalization, higher incidence of complications, or higher rate of readmissions in the first 30 days after discharge in patients with mysosteatosis.

The relationship between body composition parameters and cancer outcomes has been extensively investigated. In a large metanalysis from Aleixo et al., the presence of myosteatosis was associated with significant worse prognosis in patients with different types of neoplasia, including HCC [5]. Regarding HCC, the results are conflicting and mostly derive from patients undergoing hepatic resection. In a cohort of 100 consecutive patients recently published by Meister et al., patients with myosteatosis presented a significantly higher number of major post-operative complications and transfusion requirement compared to patients with normal muscle density [12]. Interestingly, long-term overall survival and recurrence-free survival were not different between the two groups, even if survival tended to be worse in the group with reduced muscle density. Similarly, Jang et al. failed in finding significant differences in overall survival in a cohort of 160 Korean patients with surgically treated HCC [17]. Fujiwara and colleagues described the body muscular composition of a wide cohort of patients with HCC [18] including those with advanced neoplastic disease (BCLC B or more). In that study, patients with low muscle attenuation presented with significantly higher mortality both for all causes and liver-related events. Taken altogether, these findings do not suggest a universal role of myosteatosis as a predictor of outcome in patients with HCC, but rather suggest that different patients and HCC features that might be linked to myosteatosis could, in part, explain the contradictory findings.

Among the relevant patients’ characteristics that could confound the data is the underlying liver disease etiology as the etiology of cirrhosis has been shown to be a driver in the development of myosteatosis. In our study, alcoholic and metabolic liver diseases were more frequent in the myosteatosis group, while viral liver disease were more represented in patients without muscular abnormalities. Similar findings have been reported by Meister et al. [12], while no differences regarding cirrhosis etiology were reported in other populations [2].

Sarcopenia can be ruled out as a confounder in our study, as the prevalence of sarcopenia measured by CT scan was similar in both groups (85.1% in the myosteatosis group vs. 86.1% in non-myosteatosis group). Given that sarcopenia emerged as an independent predictor of outcome, this highlights that the assessment of muscle mass or muscle composition should not be considered equivalent in assessing frailty in patients with HCC.

We do acknowledge our study has several limitations mainly due to its retrospective nature and the relatively small number of patients included. Also, our cohort of patients reflects the epidemiology of liver diseases in Italy, where viral hepatitis is the main cause of cirrhosis and liver cancer. Whether our findings are true also in areas with high prevalence of NAFLD/ALD needs validation. Lastly, we included only patients of Caucasian ethnicity, thus, our results are not transferrable to other populations.

Despite these limitations, to the best of our knowledge, this is the first study evaluating the role of myosteatosis in the prognosis of cirrhotic patients with intermediate HCC undergoing locoregional treatment. The lack of a significant impact of this variable on safety and outcome of TAE suggests against routine assessment of myosteatosis. On other hand, the observation that in the same cohort sarcopenia plays a major role on survival, calls for individualized treatment/nutritional interventions to improve muscle mass in patients with HCC.

## Figures and Tables

**Figure 1 jcm-12-00262-f001:**
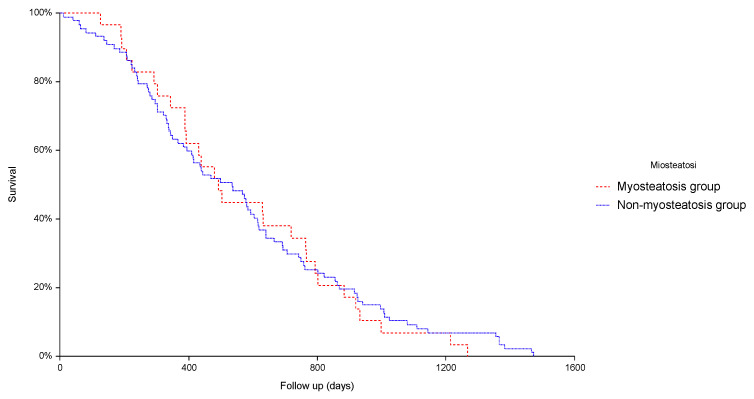
Kaplan-Meier curves of survival rates in patients with and without myosteatosis (*p* = 0.78).

**Table 1 jcm-12-00262-t001:** Main characteristics of the whole cohort and univariate analysis between patients with and without myosteatosis.

Variable	All Patients (*n* = 151)	Myosteatosis Y (*n* = 115)	Myosteatosis N (*n* = 36)	*p* Value
Age (yr)	73.2 ± 9.3	74.8 ± 7.8	68.0 ± 11.7	0.0001
Gender	M = 76.8% F = 23.2%	80.9% 19.1%	63.9% 36.1%	0.035
BMI (kg/m^2^)	25.8 ± 4.9	26.6 ± 5.1	23.4 ± 3.3	0.0006
AST (U/L)	67 ± 50	66 ± 52	71 ± 39	0.54
ALT (U/L)	57 ± 55	56 ± 58	60 ± 48	0.68
Total bilirubin (mg/dL)	1.4 ± 0.8	1.3 ± 0.7	1.6 ± 1	0.027
Serum albumin (g/dL)	3.62 ± 0.51	3.62 ± 0.53	3.62 ± 0.49	0.98
CPT	A = 82% B = 17.2% C = 0.8%	86% 12.9% 1.1%	69% 31% 0.0%	0.07
MELD score	10 ± 2.8	9.8 ± 22	10.5 ± 4.1	0.23
ALBI score	1 = 22.6% 2 = 69.3% 3 = 8.1%	23.2% 72.1% 4.7%	22.6% 61.3% 16.1%	0.11
BCLC stage	Very early = 8% Early 28.5% Intermediate 63.5%	9.5% 29.5% 61%	5.5% 22.2% 72.3%	0.46
Cirrhosis etiology	HCV = 45.3% HBV = 5.3% Alcohol = 22.7% NASH = 15.3% Other = 11.4%	43.8% 6.1% 26.3% 16.7% 7.1%	50% 2.7% 11.1% 11.1% 25.1%	0.029
Sarcopenia	Y 85.3% N 14.7%	85.1% 14.9%	86.1% 13.9%	0.88
Performance status	0 = 43.3% 1 = 46% 2 = 10.7%	41.7% 45.2% 13.1%	48.6% 48.6% 2.8%	0.22
Previous treatments	Y = 41.1% N = 58.9%	Y = 40.9% N = 59.1%	Y = 41.6% N = 58.4%	0.93
Number of nodules	Monofocal = 28% Bifocal = 22% Multifocal = 50%	28.9% 21.9% 49.2%	25% 22.2% 52.8%	0.89
Major lesion diameter (mm)	32.8 ± 22.4	33 ± 22.4	32.3 ± 22.5	0.88
Length of hospitalization (days)	2 ± 1.7	2.1 ± 1.8	1.8 ± 1.3	0.36
Complications after procedure	Y = 7.3% N = 92.7%	7.8% 92.2%	5.5% 94.5%	0.64
Rehospitalization in 30 days	Y = 6% N = 94%	6% 94%	5.5% 94.5%	0.90
Complete response	Y = 27.1% N = 72.9%	29.3% 70.7%	20% 80%	0.27
Survival (days)	808 ± 673	816 ± 682	804 ± 650	0.92

## Data Availability

The data presented in this study are available on request from the corresponding author.

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
