# Peer review of "Myosteatosis Is Not Associated with Complications or Survival in HCC Patients Undergoing Trans Arterial Embolization"

_jcm, 2022, doi:10.3390/jcm12010262_

Round 1

Reviewer 1 Report

jcm-2062844-peer-review-v1

Myosteatosis Is Not Associated with Complications or Survival in HCC Patients Undergoing Trans Arterial Embolization

This is a retrospective analysis of a cohort of 151 cirrhotic patients with unresectable hepatocellular carcinoma (HCC) who underwent their 1st trans-arterial embolization (TAE) between March 1st 2011 and July 1st 2019 in single center. Sarcopenia was assessed using the L3-SMI method while myosteatosis with a dedicated segmentation suite (3D Slicer), using a single slice at an axial plane located at L3 and calculating the IMAC (Intramuscular Adipose Tissue Content Index).

They showed that myosteatosis, fatty infiltration of the muscle assessed by CT scan, was highly prevalent (76%) in this study’s patients with HCC who received locoregional treatment. Sarcopenia prevalence was similar in both groups (85.1% in the myosteatosis group vs 86.1% in non-myosteatosis group).

The authors concluded that lack of a significant impact of myosteatosis on safety and outcome of TAE suggests against routine assessment of myosteatosis. On other hand the observation that in the same cohort sarcopenia plays a major role on survival, calls for individualized treatment/nutritional interventions to improve muscle mass in patients with HCC.

-       This is small sample size study is missing BCLC classification and AlBi score as well ECOG, which are important predictors of mortality and survival.

-       Also, the design of the study and categorizing the patients to different groups of myosteatosis  and no- myosteatosis is completely wronge. The authors shoud perform multivariable analysis.

-       It is not clear how the authors came up with the sex-specific cut-off values of IMAC > -0.44 in male and > -0.31 in female for defining myosteatosis. There is no citation to statement or there is no statistical desci[ption on who these number were calculated. We are dealing with the different group of population than normal population sand these cut-offs should be calculated specifically for this group of patients.

-       Additionally, the studied population underwent variable TAE, ranging from microparticles TAE (P-TAE), microparticles plus cyanoacrylate glue TAE (G-TAE) and Lipiodol TAE (L-TAE).

Author Response

  • Thank you for the comment. We have calculated ALBI score and BCLC in our cohort, which are added in our manuscript. Both these scores didn't result significantly associated with myosteatosis.
  • Thank you for the comment. We added a multivariate analysis, with age and BMI resulting independent predictors of myosteatosis.
  • Actually no specific guidelines are available, so we considered as cutoffs for the diagnosis of myosteatosis values which were previously used in the most important published papers in this field, which we've already cited.
  • Thank you for this observation. Different types of TAE have been used because patients were treated along different years. However, we don't think that different types of TAE may have influenced final results.

Reviewer 2 Report

Dear Authors,

 your article is very interesting, well organized  and clearly written.. Two sentences in  Design of the study  demand revision, because they are not clear.  "All patients underwent a complete disease staging before being referred to TAE and they were followed up with CT imaging 1 month after TAE. In case of disease recurrence or partial response to treatment, patients were re-discussed; otherwise, a 3-month follow-up interval was established."

Author Response

Thank you for the comment. We changed the sentences as follows: All patients underwent a CT scan before being referred to TAE and they were followed up with CT imaging 1 month after the procedure. In case of disease recurrence or partial response to treatment, patients were re-discussed with a multidisciplinary team to decide the subsequent treatment; otherwise, a 3-month follow-up interval was established.